# Copeptin in Patients with Pregnancy-Induced Hypertension

**DOI:** 10.3390/ijerph18126470

**Published:** 2021-06-15

**Authors:** Agnieszka Marek, Rafał Stojko, Agnieszka Drosdzol-Cop

**Affiliations:** Chair and Department of Gynecology, Obstetrics and Oncological Gynecology, Medical University of Silesia in Katowice, Markiefki 87, 40-211 Katowice, Poland; rafal@czkstojko.pl (R.S.); cor111@poczta.onet.pl (A.D.-C.)

**Keywords:** pregnancy inducted hypertension, copeptin, vasopressin

## Abstract

Pregnancy-induced hypertension (PIH) occurs in 6–8% of pregnancies, and increases the risk of many severe obstetric complications. The etiology of PIH has not been fully explained, and hence, treatment is only palliative in nature, and prevention is not fully effective. It has been proposed that PIH development is influenced by the arginine vasopressin pathway, whose surrogate biomarker is copeptin. The aim of this study is a prospective assessment of the relationship between the level of copeptin in pregnant women and the occurrence of PIH, and to identify its usefulness in predicting complications. The study involved a group of 21 pregnant women who developed PIH and 37 women with uncomplicated pregnancies as a control group. Blood samples were collected at the three trimesters of gestation (<13 HBD, between 13 and 26 and >26 HBD) and then frozen. Copeptin levels [pg/mL] were measured in serum samples obtained in the first, second and third trimesters of gestation from women in the PIH and control groups. The concentration of copeptin in the second and third trimesters of pregnancy was statistically significantly higher in the PIH group (*p* < 0.05). For copeptin determined in the first trimester, which could be used to screen for PIH, the area under the ROC curve was 0.650. The highest risk of PIH occurred in patients with high concentrations of copeptin in the first trimester of pregnancy and obesity OR = 5.5 (95% CI 1.0–31.3). The risk of PIH was augmented in patients with high levels of copeptin and an abnormal Doppler result of the uterine arteries OR = 28.4 (95% CI 5.3–152). In conclusion, copeptin levels were found to be elevated in pregnant women before the diagnosis of PIH; however, copeptin should not be used as a stand-alone marker. The combination of copeptin concentration with the other risk factors (diabetes, maternal age and preeclampsia in previous pregnancy) did not improve the diagnostic values of the use of copeptin in the PIH risk assessment, but the combination of copeptin concentration with BMI may be useful in clinical practice. Measurement of copeptin together with a Doppler examination of uterine arteries in the first trimester of pregnancy may be a useful marker in predicting the development of PIH.

## 1. Introduction

Pregnancy-induced hypertension (PIH) complicates 6–8% of pregnancies and is defined as arterial hypertension (≥140/90 mmHg) that occurs after the 20th week of pregnancy along with all the puerperium [1,2]. High blood pressure during pregnancy increases the risk of many obstetric complications.

The risk factors for pregnancy-induced hypertension are: the presence of PIH in previous pregnancies and any occurrence in the family history, age ≥ 40 years, obesity and chronic maternal diseases (kidney diseases, diabetes, autoimmune disorders) [3].

Arterial hypertension in pregnancy increases the risk of many obstetric complications: fetal growth restriction (FGR), prematurity with all its consequences, intrauterine death, premature separation of the placenta and intrauterine hypoxia [4]. Pre-eclampsia and eclampsia can lead to renal and circulatory failure, hypertensive encephalopathy, bleeding into the central nervous system and disseminated intravascular coagulation (DIC), and is estimated to cause 14% of deaths in pregnant women worldwide [5]. As a result, the costs of health care for pregnant woman and the outlays related to prematurity increase significantly.

Due to the unexplained etiology of pregnancy-induced hypertension, the prevention and treatment regime of this complication are not effective enough [6]. That is why it is so crucial to reveal a marker that allows us to predict the occurrence of PIH. One possible candidate may be the arginine vasopressin pathway.

It is believed that vasopressin plays an indispensable role in the pathogenesis of pregnancy-induced hypertension [7,8,9]. In the course of PIH, endothelial and vascular dysfunction, immune and angiogenesis disorders, as well as oxidative stress, are observed [10]. The influence of vasopressin on all the above-mentioned processes is known, hence the hypothesis that its excessive secretion may be a factor that initiates the development of PIH.

The half-life of an antidiuretic hormone is only 23 min, which makes it very intractable to determine. Copeptin is the c-terminal part of preprovasopressin, from which vasopressin is also formed. The determination of its concentration as a more stable protein indirectly proves the quantity of the synthesized antidiuretic hormone [11].

Analyzing the above data and the results of studies on the concentration of copeptin in patients with developed preeclampsia, an experiment was designed to compare the concentration of circulating copeptin in the blood of pregnant women in each trimester who developed pregnancy-induced hypertension and in those who without such complications.

The main aim of the investigation was to assess the usefulness of copeptin level measurement as a biomarker for predicting pregnancy-induced hypertension.

## 2. Materials and Methods

The study covered a population of 350 patients of the Gynecology and Obstetrics Ward with the Gynecological Oncology Subdivision of the Hospital of the Order of Bonifraters in Katowice, the Gynecology and Obstetrics Clinic of the Bonifraters’ Health Center and other obstetrics clinics in Katowice in the period from June 2017 to May 2019.

Twenty-one pregnant women, invited to participate in the study and who were observed from the beginning of pregnancy and developed PIH, were eligible to join the study group (hereinafter referred to as the PIH group). The criteria for inclusion in the study group were: single pregnancy confirmed by ultrasound, obtained free and informed consent of the patient, age ≥ 18 years, the occurrence of hypertension (≥140/90 mmHg) after the 20th week of pregnancy, likewise in the puerperium, and performance of the blood test for PAPP-A and USG Doppler of the uterine arteries in the first trimester. The criteria for exclusion from the study group were: chronic hypertension on the basis of a detailed medical interview and the patient’s medical records, the occurrence of chromosomal aberrations and structural malformations in the fetus, multiple pregnancy, lack of informed and free consent of the patient, age < 18 years, failure to perform the blood test for PAPP-A and USG Doppler of the uterine arteries in the first trimester, and patient’s unwillingness to cooperate.

Thirty-seven healthy pregnant women whom we observed since the beginning of pregnancy were eligible to join the control group. The criteria for inclusion in the control group were: single pregnancy confirmed by ultrasound, obtained free and informed consent of the patient, age ≥ 18 years, normal blood pressure values during pregnancy, performance of the blood test for PAPP-A and USG Doppler of the uterine arteries in the first trimester. The exclusion criteria from the control group were: finding, on the basis of a detailed medical interview and the patient’s medical records, of chronic arterial hypertension, pre-pregnancy diabetes, chronic kidney diseases and autoimmune diseases in a pregnant woman, the occurrence of chromosomal aberrations and structural malformations in the fetus, multiple pregnancy, lack of informed and free consent of the patient, age < 18 years, failure to perform the blood test for PAPP-A and USG Doppler of the uterine arteries in the first trimester, and patient’s unwillingness to cooperate.

The study was approved by the Bioethical Committee of the Medical University of Silesia in Katowice (decision No. KNW/0022/KB1/55/III/16/17 of 3 October 2017).

All study participants were asked to complete a questionnaire containing basic demographic and medical data at the time of their first blood sample collection. The information was updated during subsequent visits in the second and third trimesters of pregnancy as well as after delivery and in the postpartum period.

Obesity in pregnancy was defined as a maternal BMI ≥ 30 at the antenatal booking visit. A patient who performed 30 min of moderate exercise 3 times a week was considered physically active. Prophylaxis with acetylsalicylic acid was considered to be taking 75–150 mg of ASA once a day from the 16th week of pregnancy or earlier.

Gestational diabetes mellitus (GDM) was defined as any degree of glucose intolerance with onset or first recognition during pregnancy. A fasting plasma glucose level > 126 mg/dL (7.0 mmol/L) or a casual plasma glucose >200 mg/dL (11.1 mmol/L) have meet the threshold for the diagnosis of diabetes, if confirmed on a subsequent day, and precludes the need for any glucose challenge. Fasting glucose measurement or diagnostic OGTT were performed during the first visit. The glucose threshold value was ≥93 mg/dL in the fasting state or ≥180 mg/dL after 1 h or ≥153 mg/dL after 2 h. A normal OGTT result at the beginning of pregnancy was an indication to repeat the test between 24 and 28 weeks of pregnancy [12].

Blood pressure was measured while patient was seated, with feet supported, for 2–3 min before blood pressure is measured. Blood pressure was taken on both arms. The right arm was used thereafter if there is no significant difference between the arms; otherwise, the higher values was used. A standard cuff was used for arms with a circumference of ≤33 cm while the large cuff (15 × 33 cm bladder) was used for arms with a circumference of >33 cm. Certified electrical pressure gauges were used. The prerequisite for the diagnosis of pregnancy-induced hypertension was the presence of elevated blood pressure (≥140/90 mmHg) in two independent measurements 6 h apart [13].

Each participant had a prenatal test at between 11^+0^ and 13^+6^ weeks of pregnancy. The description of the ultrasound examination in terms of multiple pregnancies and fetal malformation was used. The results of the PAPPA protein determination were also used in the study.

Uterine artery Doppler flow was performed in pregnant women during the 11^+0^ and 13^+6^ week of gestation. All scans were performed by experienced sonographers (RMS) using transvaginal (5 to 9 MHz) or transabdominal (4 to 8 MHz) probe. The Doppler flow was analysed according to existing guidelines (ISUOG). The signal was updated until at least three similar consecutive waveforms were obtained, and both the pulsatility index (PI) and resistance index (RI) of the right and left UA was measured electronically once and recorded as well as the presence and/or absence of protodiastolic notch. The mean PI of the left and right uterine arteries was taken into account in the study [14].

In the present research, blood samples were collected at three time intervals (<13 weeks gestation, between 13th and 26th weeks gestation, and >26th weeks gestation). Blood was collected in clot tubes. The blood was then centrifuged and subsequently frozen at −20 °C for copeptin concentration measurement, which was performed systematically thereafter. Analysis of copeptin levels was performed in all three blood samples of women diagnosed with pregnancy-induced hypertension and in a similar numerical and demographic group of women whose pregnancy was uneventful.

The relative measurement of copeptin concentration was performed by the ELISA method in stored plasma samples with the use of a ready-made Copeptin reagent kit (USCN, Houston, TX, USA) in the laboratory of the Department of Pathophysiology of the Medical University of Silesia. The sensitivity of the method is 5.7 pg/mL.

In our statistical analysis, the Excel 2007 (Microsoft, Redmond, WA, USA) and STATISTICA10 software (StatSoft, Cracow, Poland) were used. The result of the statistical test was considered statistically significant if the obtained significance level *p* was less than or equal to 0.05. The statistical research used: Kolmogorov-Smirnov distribution normality test; Levene’s test of homogeneity of variance; Student’s *t*-test; Fisher’s exact test; Pearson’s correlation test; ROC (receiver operating characteristic) analyzes; Mc Nemar’s test.

The distributions of these variables were not statistically significantly different from the normal distribution (Kolmogorov-Smirnow test). Student’s *t*-tests were used to compare the mean differences between the two groups. In the case of homogeneity of variance (Levene’s test), the main version of the Student’s *t*-test was used, and for a significant lack of homogeneity-the Student’s *t*-test with independent variance estimation. The correlation of two quantitative variables was based on the Pearson test.

ROC (receiver operating characteristic) curves were used to evaluate the diagnostic tests. ROC curves are built to assess the quality of a diagnostic test based on the values of the chosen quantity, e.g., copeptin. The ROC curve is determined in such a way that for the next, changing value (in the entire range of its occurrence), the presence of the fractions of true positive and false positive results may be determined. For the curve determined in this way, the area underneath it (AUC) is an assessment of the quality of the test. The closer this field value is to one, the better the diagnostic test. The minimum value for the diagnostic test is 0.500. A value of 1000 is ideal. In practice, the AUC ranges between 0.800 and 0.950. The optimal value of the diagnostic test cut-off point, based on the ROC curve, is the value for which the point on the ROC curve is closest to the point (0%; 100%) of the graph.

The Mc Nemar test for the adopted cut-off examines whether the division made by the diagnostic test in the group of all scores is the same as that made by clinical evaluation (control subjects) or another clinical reference test. In the case of the MC Nemar test result (not statistically significant difference), the results of the diagnostic test are consistent with the division made by clinical evaluation or other clinical reference test.

## 3. Results

### 3.1. Cohort Characteristics

There were no significant statistical differences between the two groups in age and weight gain. PIH group had statistically significantly higher body weight (78.4 ± 19.0 vs. 65.2 ± 13.0, *p* = 0.003) and BMI at the beginning of pregnancy (28.9 ± 7.7 vs. 23.6 ± 4.3, *p* = 0.008). Also, the values of body weight at the end of pregnancy (93.8 ± 17.0 vs. 76.9 ± 12.7, *p* = 0.0007) and BMI in this period (34.6 ± 6.9 vs. 27.8 ± 4.2, *p* = 0.00002) were statistically significantly higher in PIH group. There were no statistically significant differences between the two groups in terms of physical activity levels, prophylaxis with acetylsalicylic acid and the occurrence of gestational diabetes (GDM) in the observed pregnancy (*p* > 0.05). There was also no statistically significant difference in the percentage of the primiparas and the obstetric history with pregnancy-induced hypertension and fetal growth restriction between the control and PIH groups (*p* > 0.05). Table 1 presents the general characteristics of the study population.

### 3.2. Concentrations of Copeptin in the Blood Serum in the Study and Control Groups

In the study, no statistically significant differences were found in the concentrations of copeptin in the first trimester of pregnancy between pregnant women who developed PIH and the group of women experiencing normal pregnancies (69.5 ± 38.2 vs. 51.5 ± 31.5, *p* = 0.08). On the other hand, the concentration of copeptin in the second and third trimesters of pregnancy was significantly higher in the PIH group (96.9 ± 65.6 vs. 59.1 ± 34.6, respectively, *p* = 0.02 and 106.9 ± 68.5 vs. 66.0 ± 40.7, *p* = 0.02). Figure 1 shows the values of copeptin in the PIH group and in the control group in subsequent trimesters of pregnancy.

The week of pregnancy in which blood samples were collected in the first and second trimesters did not differ significantly between the groups. However, significant differences were observed in the third trimester, when blood samples were taken earlier in women from the PIH group (35.5 ± 3.2 vs. 38.0 ± 2.8, *p* = 0.003), which can be explained by indications for the earlier termination of pregnancy in this group of patients.

### 3.3. Correlations between Copeptin Concentrations and Demographic Factors, Obstetric Results and Blood Pressure Values

There was no statistically significant relationship between the concentration of copeptin and maternal age, BMI at the beginning of pregnancy or duration of pregnancy in the PIH and control groups (*p* > 0.05). However, in the control group, a correlation was found between the concentration of copeptin during the entire pregnancy and the weight of the newborn (r = 0.35, *p* < 0.05). In the study and control groups, there was also no relationship between the concentration of copeptin and fertility, the type of delivery, physical activity during pregnancy, the occurrence of GDM in the current pregnancy, and a history of PIH and FGR.

In the control and PIH groups, no noteworthy reciprocity was found between the concentration of copeptin and the mean arterial pressure and PI of the uterine artery (*p* > 0.05). On the other hand, the inverse correlation was shown between the concentration of PAPP-A protein taken in the first trimester and the concentration of copeptin in the second trimester in the PIH group (r = 0.45, *p* = 0.04).

### 3.4. The Use of Copeptin in Predicting the Development of Pregnancy-Induced Hypertension

Analyzing the quality of copeptin as a test to predict the development of pregnancy-induced hypertension, the results showed undesirable AUC values at each stage of pregnancy. In practice, tests are used for which the AUC is between 0.800 and 0.950. For copeptin measured in the first trimester, which could be used in screening of patients at risk of developing PIH, and by to implement prophylaxis with acetylsalicylic acid, the area under the ROC curve was only 0.650. ROC charts of a diagnostic test for PIH based on the measurement of copeptin concentration in subsequent trimesters of pregnancy are presented in Figure 2. Using a first trimester copeptin concentration of 55 pg/mL as the cut-off point, a sensitivity of 67% (95% CI 43–86%) and a specificity of 62% (95% CI 44–78%) in predicting PIH were obtained. The odds ratio for copeptin in the first trimester was 3.3 (95% CI 1.1–10.1). The highest values of the diagnostic test measures were obtained for the concentrations of copeptin in the third trimester.

### 3.5. The Use of Copeptin and Demographic Data in Predicting the Development of Pregnancy-Induced Hypertension

In the conducted study, the high risk factors for the development of pregnancy-induced hypertension included obesity OR = 5.1 (95% CI 1.3–19.8), diabetes OR = 3.3 (95% CI 0.8–13.4) and the presence of PIH in the previous pregnancy, OR = 3.9 (95% 0.5–28.2). Using the measurements of copeptin concentrations in the first trimester and the presence of the high risk factors presented above, it could be concluded that the highest risk of PIH occurred in patients with high copeptin concentration and obesity OR = 5.5 (95% CI 1.0–31.3). Increased risk also occurred in pregnant women with high levels of copeptin and diabetes during pregnancy OR = 1.8 (95% CI 0.2–14.1) and older than 35 years OR = 1.2 (95% CI 0.2–7.8). These tests were characterized by high specificity (95%) but low sensitivity (10%).

### 3.6. Use of Copeptin and Measurement of Mean Arterial Pressure, PAPP-A Protein Concentration and PI of Uterine Arteries in the First Trimester of Pregnancy to Predict the Development of Pregnancy-Induced Hypertension

Much higher values of sensitivity and specificity occurred when the combination of high values of copeptin concentrations in the first trimester with mean arterial pressure (sensitivity = 52%, specificity = 92%), PAPP-A protein concentration (sensitivity = 48%, specificity = 89%) and PI in uterine arteries (sensitivity = 62%, specificity = 96%) was used in diagnostics. The risk of pregnancy-induced hypertension was highest in patients with high copeptin levels and abnormal uterine Doppler results OR = 28.4 (95% CI 5.3–152). There was also a high risk in women with abnormal MAP OR values = 12.5 (95% CI 2.9–53.6) and a high concentration of PAPP-A OR = 7.5 (2.0–28.8).

## 4. Discussion

The main limitation in the study is the limited number of studies that can be cited in the discussion and the great diversity of experimental approaches.

In the conducted study, the concentration of copeptin was significantly higher in the second and third trimesters of pregnancy in women who developed PIH compared to those experiencing uncomplicated pregnancies. Our study showed that copeptin is characterized by low sensitivity and specificity in predicting the development of pregnancy-induced hypertension, and therefore, that it should not be used as a prospective marker. Several studies support the use of copeptin assays as a new biomarker of preeclampsia in early pregnancy. Santillan et al. measured the concentration of copeptin in the first, second and third trimesters of pregnancy in 50 pregnant women who developed preeclampsia, 54 patients experiencing normal pregnancies and 33 nonpregnant women. Compared to our study, much higher concentrations of copeptin were revealed in each trimester in patients with complicated pregnancies compared to patients with physiological and nonpregnant pregnancies. The determination of copeptin concentration in the prediction of preeclampsia is characterized by high sensitivity, specificity and a high positive and negative predictive value [15]. Tuten et al. studied the concentration of copeptin in patients with early and late preeclampsia. Blood samples were taken once around 30 weeks of gestation. The mean values of copeptin concentration in both groups with preeclampsia were higher compared to the control group, which were healthy pregnant women. This difference was statistically significant only in the group of pregnant women with early preeclampsia [16]. In 2020, Bellos et al. conducted a meta-analysis examining the association between serum copeptin levels and preeclampsia risk. Their findings suggested that preeclampsia is associated with significantly elevated serum copeptin levels, irrespective of disease severity and onset. Women that developed preeclampsia presented increased copeptin concentration during all pregnancy trimesters, indicating that an increase in its values may precede the clinical manifestation of the disease [17]. Based on the results of presented studies, it can be concluded that an increased maternal copeptin level may be associated with the etiology of preeclampsia, and may be used to predict its severity. Various studies on the concentration of copeptin in patients with PE differed significantly in methodology, e.g., in some, the concentration of copeptin was measured after the diagnosis of hypertension [18,19,20]. It should be noted that the exact population and timing for copeptin evaluation in order to achieve optimal predictive efficacy remains to be elucidated. The findings of our and cited studies should be confirmed by future large-scale prospective cohort studies in order to introduce cut-off values that would enable the early detection of the disease.

Just like in our study, the increase in the concentration of copeptin also occurred in the course of physiological pregnancy, but mainly in pregnancies complicated by preeclampsia. Few studies have investigated the concentration of copeptin in patients with PIH. Young et al. studied the concentration of copeptin in the course of various complications of pregnancy: preeclampsia, gestational diabetes, PIH and preterm labor [21]. In a study by Young et al., elevated copeptin levels were confirmed in women before preeclampsia was diagnosed. No reliable relationships were found between the concentration of copeptin and the development of PIH, gestational diabetes, proteinuria and preterm labor, or not in PIH alone. The results of this study suggest that copeptin is a marker which is specific to the development of preeclampsia. In our study, we found that elevated copeptin levels also occurred prior to the development of PIH. Okwor et al. conducted a case-control study comprising 156 pregnant women grouped into those with chronic hypertension (CH), gestational hypertension (GH), and preeclampsia (PE) as cases and normotensives as controls [22]. Serum copeptin and plasma BNP levels were measured. Mean serum copeptin and plasma BNP were significantly higher in women with GH and PE compared with controls (*p* < 0.05). Researchers found that serum copeptin and plasma BNP may be used as markers of hypertensive disorders of pregnancy. According to our study, copeptin should not be used to predict PIH as a standalone marker. More research is needed to determine the use of copeptin in the development of PIH.

In our study, no statistically significant correlation was found between the concentration of copeptin and the mother’s age, BMI value at the beginning of pregnancy and the duration of pregnancy in the control group and the PIH group. There were no significantly higher concentrations of copeptin in patients who experienced PIH or IUGR in a previous pregnancy and primiparas. There are few reports in the literature on the correlation of copeptin concentrations with other risk factors for preeclampsia. In the study by Young et al., the levels of copeptin were remarkably higher (*p* < 0.05) in the group of younger, black, unmarried and low-material patients. In our study, we did not collect information on the material and marital status of patients. The quoted study also proved, as previously mentioned, that the coexistence of gestational diabetes and PE did not significantly increase the concentration of copeptin compared to the group with isolated preeclampsia [21]. Tuten et al. showed that the levels of copeptin correlate only with gestational age and blood pressure. The relationship between copeptin concentrations and age, as well as maternal BMI, newborn birth weight and PI of the uterine arteries, has not been confirmed [16]. More research is needed to determine the use of copeptin and PIH risk factors in predicting PIH development.

In our study, it was not possible to prove the relationship between the concentration of copeptin and the PI of the uterine arteries. Yesil et al. designed a study to use Doppler imaging of the uterine arteries and copeptin to identify patients at high risk of developing preeclampsia. The level of copeptin was higher in women with abnormal results of Doppler examinations of the uterine arteries between 20 and 24 + 6 weeks of gestation (*p* = 0.001) [23]. Zulfikaroglu et al. showed that the concentrations of copeptin were significantly higher in PE patients with abnormal Doppler results compared to the group of patients with preeclampsia and normal uterine artery flow [20]. Further research on this topic is necessary because of the small number of reports on the correlation of copeptin with results of Doppler examination of the uterine arteries for preeclampsia.

## 5. Conclusions

The obtained results show that in the group of patients with pregnancy-induced hypertension, the concentration of copeptin was increased from the second trimester of pregnancy. It has been shown that copeptin, as a standalone marker, should not be used in predicting PIH. On the other hand, the combination of copeptin with the Doppler examination of the uterine arteries in the first trimester of pregnancy is characterized by high sensitivity and specificity, which can be used in the screening of pregnancy-induced hypertension. The presented research work does not explain the pathophysiological mechanism of PIH. It should be noted that the sample size was limited, so the results are preliminary.

It seems justified to conduct further research on the role of the arginine vasopressin pathway in the etiology of PIH.

## Figures and Tables

**Figure 1 ijerph-18-06470-f001:**
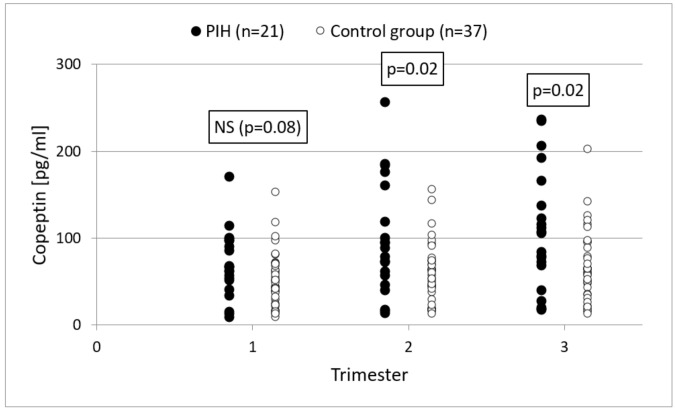
The values of copeptin in the PIH group and in the control group in subsequent trimesters of pregnancy.

**Figure 2 ijerph-18-06470-f002:**
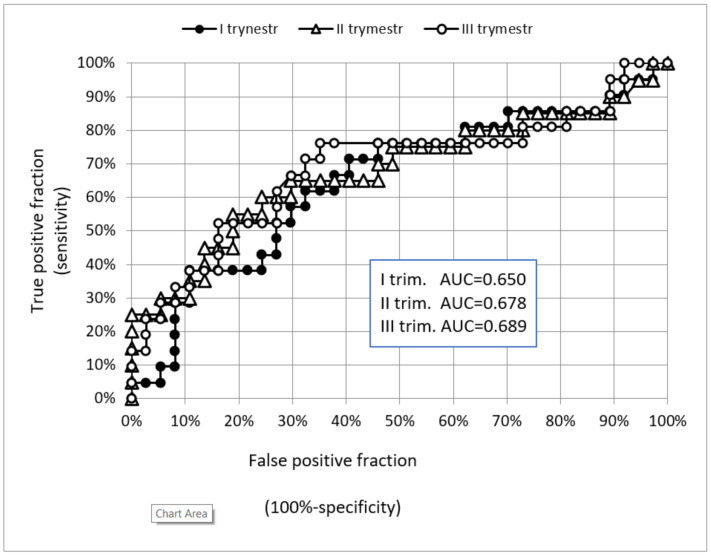
ROC (receiver operating characteristic only curve) for copeptin as a diagnostic test for PIH. Measurement of copeptin concentration was done in subsequent trimesters of pregnancy.

**Table 1 ijerph-18-06470-t001:** Group characteristics (^a/^ Student’s *t*-test; ^b/^ Fisher’s exact test).

	Study Group PIH (*n* = 21)	Control Group K (*n* = 37)	PIH vs. K
Age at birth [years]	29.4 ± 5.1	29.9 ± 5.0	NS (*p* = 0.72) ^a/^
Starting body weight [kg]	78.4 ± 19.0	65.2 ± 13.0	*p* = 0.003 ^a/^
Starting BMI	28.9 ± 7.7	23.6 ± 4.3	*p* = 0.008 ^a/^
Final body weight [kg]	93.8 ± 17.0	76.9 ± 12.7	*p* = 0.00007 ^a/^
Final BMI	34.6 ± 6.9	27.8 ± 4.2	*p* = 0.00002 ^a/^
Weight gain [kg]	15.5 ± 7.0	12.2 ± 4.8	NS (*p* = 0.06) ^a/^
Obesity	8 (38.10%)	4 (10.81%)	*p* = 0.02 ^b/^
Nicotynism	1 (4.76%)	1 (2.70%)	NS (*p* = 0.99) ^b/^
Physical activity	8 (42.86%)	23 (62.16%)	NS (*p* = 0.18) ^b/^
PIH in previous pregnancy	3 (30.00%) (for *n* = 10)	2 (10.00%) (for *n* = 20)	NS (*p* = 0.30) ^b/^
FGR in previous pregnancy	1 (10.00%) (for *n* = 10)	1 (5.00%) (for *n* = 20)	NS (*p* = 0.99) ^b/^
GDM	4 (19.05%)	4 (10.81%)	NS (*p* = 0.44) ^b/^
Primiparas	11 (52.38%)	15 (40.54%)	NS (*p* = 0.42) ^b/^

## Data Availability

Data is contained within the article.

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
