# Peer review of "Copeptin in Patients with Pregnancy-Induced Hypertension"

_ijerph, 2021, doi:10.3390/ijerph18126470_

Round 1
Reviewer 1 Report
The study is original because there are few studies that relate copeptin to PIH.
1.-In the study design is mentioned that it is a prospective study when the samples were actually retrospectively analyzed once the pregnant women developed PIH and delivery occurred. First you got the blood samples and afterwards you selected the groups depending on the obstetric outcomes.This should be better explained in material and methods.
2.- How were the pregnant women in the control group selected? Why are there 37? Was an attempt to choose a homogeneous group in relation to age, time of delivery, etc.?
3.- The definition of PIH should include arterial hypertension after the 20th week of pregnancy.
4.- The results should be adjusted with the BMI because there are significant differences between both groups and this can be a confounding factor and we know that pre-pregnancy obesity is a risk factor for PIH. Multivariate analysis should be done.
5.-Only studies with preeclampsia appear in the discussion. Any study comparing copeptin and PIH should be mentioned. The authors did not evaluate the pregnant women who developed PE?
6.- If copeptin has low diagnostic sensitivity and specificity, it does not seem to be a good biomarker for PIH.
Reviewer 2 Report
This is a quite interesting study on the predictive role of copeptin as a biomarker for early detection of pregnancy-inducted hypertension. Nevertheless, the document does not meet the formal criteria of an original manuscript, and all sections need deep revision and improving, especially the Material and Methods and the Discussion sections. The Results section is not written according to scientific language, nor in past tense; besides, an objective description of the results is usually lacking. Finally, some information is in a wrong section.
As a general comment, the Material and Methods section is incomplete, and it does not contain a clear description of diagnostic procedures and statistical analysis of data. The Discussion section must be revised and modified since it does not adjust to the standard format required for this section. Moreover, authors must revise and improve English grammar.
Specific comments.
- Line 14. “While the research project”. What does it mean?
- Line 16. Please, change “of the level of copeptin was performed” by “copeptin levels were measured”.
- Line 16. “in all serum samples”. What do the authors refer? To serum samples obtained in the first, second and third trimester of gestation? If so, please rewrite the sentence and provide a clear message.
- Lines 17-18. Please, provide P values. What the study group is? The PIH group? If so, please indicate.
- Line 19. What is the point of this clarification?
- Line 20. Please change “is” by “was” and “occurs” by “occurred”.
- Line 21. How was the obesity diagnosed? How was the OR calculated? Please, indicate briefly.
- Line 22. Please, change “is” by “was”.
- Line 23. Please, change “Summarizing” by “In conclusion”.
- Line 26. Please change “does not” by “did not”.
- Lines 25-27. The content of these line does not agree with the information given in previous lines. Please, revise and correct it.
- Line 27. How and when the Doppler examination of uterine arteries was performed? Please, proved a detailed explanation in the Material and Methods section.
- Lines 33-36. Please, revise these lines since they contain identical words as lines 7-8.
- Lines 48-49. Please, provide a reference at the end of the sentence.
- Lines 58-61. Please, provide references supporting this information.
- Lines 62-63. Authors must improve this sentence. Which “above data” were analysed? Which studies?
- Line 68. Please, change “level” by “levels”.
- Line 69. “test for the worthwhile exploit of its determination”. Please, use other words.
- According to the line 72 the study covered a population of 350 patients but, according to the information given in line 76 the study group consists of 21 pregnant women and in line 89 the authors indicate that the control is of 37 people. Authors must carefully revise the information given in these lines and provide a clear information about the number of patients included in the study, as well as the number of patients per group. What is the study group? The patients’ group with PIH? If so, please refer as the PIH group throughout the whole document.
- In the Material and Method section, authors must describe the procedure followed to statistically analyse the results obtained as wells as to evaluate the correlation between variables. Did the authors check the normality and homoscedasticity of the data? How were the AUC curve and OR values obtained?
- In the Material and Method section, authors must describe how the Doppler examination was performed.
- In the Results section (lines 191-193) authors refer to obesity and diabetes? How were they diagnosed? Please, specify in the Material and Methods section.
- Lines 92-94. How these parameters were measured. Please, add a description of the procedure followed.
- Table 1 and the content of lines 103-113 must be placed at the beginning of the Results section, subsection Cohort characteristics.
- Lines 109-110. “physical active patients alike, prophylaxis with acetyl salicylic acid and the occurrence of gestational diabetes (GDM) in the observed pregnancy”. Please, describe the procedure used to measure all these variables in the Material and Methods section.
- Line 138. “no statistical meaningful difference”. This is not a proper way to describe the results obtained. Please, improve this sentence.
- Lines 143-144. “Figure 1. shows”. Please, delete the point.
- Lines 149-150. Please, provide an appropriate legend for Figure 1. Authors must indicate the meaning of PIH and the sample size of each group.
- Line 151. “Pronouncedly”. This is not an appropriate term. Please, indicate if the differences are either significant or not.
- Line 151. Please correct “were shown” by “were observed”.
- Line 152. “fundamental dissimilarity differences”. Please, eliminate this expression and indicate if the differences are either significant or not.
- Line 154. “often related”. This is quite ambiguous; authors must concretize and give a clear message based on the statistical analyses.
- Lines 158-160. This sentence is very ambiguous. Please, provide a clear message and give P values.
- Lines 160-162. Please, provide r P values confirming such a correlation. Moreover, authors must describe the procedure followed to analyse the correlation between variables in the Material and Methods section.
- Line 166. “noteworthy reciprocity”. What do the authors refer? Please, correct the whole sentence based on the results obtained from statistical analysis.
- Line 168. “inverse correlation”. Please, indicate r and P values.
- Line 173. “the results showed undesirable AUC values”. This is too vague, please provide a clear information based on both the results obtained and statistics.
- Line 177. Please, change “is” by “was”.
- Line 181. Correct “are” by “were”, and “is” by “was”.
- Authors must consider reviewing the whole Results section and use the past form.
- Lines 183-185. The content of this lines must be placed in the Conclusion section.
- Figure 2. Please correct AUC values by using a “point” instead of a “coma”.
- Lines 187-188. Please, provide a more detailed explanation of the information given in Figure 2.
- Line 191. “According to the study”. Which study do the authors refer?
- Lines 191-193. “the presence of PIH in the previous pregnancy”. This is quite confusing considering that the authors did not refer to “previous pregnancies” of the patients included in the study. Therefore, a deep revision is much warranted.
- Lines 194-196. The content of these lines must be placed in the Conclusions section.
- Line 198. Please, change “occurs” by “occurred”.
- Line 199. Please, change “are” by “were”.
- Line 200. Please change “does not” by “did not”.
- Authors must revise the whole Result section and provide a clear message. Moreover, this section must be written in past tense.
- Line 205. “Much higher values”. What do the authors refer? Please, provide a clear message by giving objective data.
- Line 208. Please, change “is” by “was”.
- Lines 209-211. The content of these lines must be placed in the Conclusions section.
- Lines 216-225. Subsection 4.1. Summary. I can not see the relevance of this subsection considering that most of the information it contains is already given in the Introduction section.
- Lines 227-310. Subsection 4.2. Comparison with Existing Literature. Please, eliminate this subsection title. Moreover, authors must revise the information given in these lines by analysing and comparing their results with those obtained in similar approaches instead of summarizing the research conducted in other studies.
- Lines 317-320. This is not a limitation of your study. Authors must start the Discussion section by highlighting the low number of studies related to this subject and the great diversity of experimental approaches.
- Lines 321-333. Subsection 4.4. Implications for Research and Clinical Practice. The information given in this section is a mere repetition of the information given in previous sections of the manuscript.
- Lines 335-341. The content of these lines does not provide any new information. This section must contain the most relevant findings of the research conducted.
- Lines 342-349. Please, avoid repeating literally the results obtained, and use new words. Authors must highlight that the sample size is reduced and so their results are preliminary.
- The reference list is quite dated. Authors must cite recent publication to support their findings.
Author Response
Response to Reviewer 2 Comments
This is a quite interesting study on the predictive role of copeptin as a biomarker for early detection of pregnancy-inducted hypertension. Nevertheless, the document does not meet the formal criteria of an original manuscript, and all sections need deep revision and improving, especially the Material and Methods and the Discussion sections. The Results section is not written according to scientific language, nor in past tense; besides, an objective description of the results is usually lacking. Finally, some information is in a wrong section.
As a general comment, the Material and Methods section is incomplete, and it does not contain a clear description of diagnostic procedures and statistical analysis of data. The Discussion section must be revised and modified since it does not adjust to the standard format required for this section. Moreover, authors must revise and improve English grammar.
Response: We would like to thank this reviewer for the critical evaluation of our manuscript. Pointing out the important aspects we missed has helped us greatly in revising the manuscript. The work has been linguistically checked by another native speaker. Very much appreciated.
Specific comments.
- Line 14. “While the research project”. What does it mean?
Response: Thanks for the suggestion. The sentence has been corrected to make it clearer.
- Line 16. Please, change “of the level of copeptin was performed” by “copeptin levels were measured”.
Response: Done. Thank you.
- Line 16. “in all serum samples”. What do the authors refer? To serum samples obtained in the first, second and third trimester of gestation? If so, please rewrite the sentence and provide a clear message.
Response: This is a good suggestion. The sentence has been corrected to make it clearer.
- Lines 17-18. Please, provide P values. What the study group is? The PIH group? If so, please indicate.
Response: The p value was given. It was specified that the study group was the PIH group. Thank you.
- Line 19. What is the point of this clarification?
Response: This explanation was intended for a group of readers with no specific knowledge of perinatology. It emphasizes that the most clinically useful screening tests are performed in the first trimester, because in the case of an increased risk of PIH, prophylaxis with acetylselicylic acid can be implemented before the 16th week of pregnancy, according to the recommendations.
- Line 20. Please change “is” by “was” and “occurs” by “occurred”.
Response: Done. Thank you.
- Line 21. How was the obesity diagnosed? How was the OR calculated? Please, indicate briefly.
Response: Information on how obesity was diagnosed was included in Material and Methods section.
The odds ratio was calculated from the formula: OR=
a=the number of people exposed to the disease agent who are sick (study group),
b=the number of people exposed to the disease agent who are not sick (control group),
c=number of people not exposed to the disease agent who are sick (study group),
d=number of non-exposed people who are not sick (control group).
Standard error of the odds ratio: SE = EXP [(1/a + 1/b + 1/c + 1/d)0,5]
Confidence limits: 95% CI = OR ± 1,96*SE
- Line 22. Please, change “is” by “was”.
Response: Done. Thank you.
- Line 23. Please, change “Summarizing” by “In conclusion”.
Response: Done. Thank you.
- Line 26. Please change “does not” by “did not”.
Response: Done. Thank you.
- Lines 25-27. The content of these line does not agree with the information given in previous lines. Please, revise and correct it.
Response: Information was revised and corrected. Thanks for the suggestion.
- Line 27. How and when the Doppler examination of uterine arteries was performed? Please, proved a detailed explanation in the Material and Methods section.
Response: Information on how and when the Doppler examination of uterine arteries was performer, has been added in the Material and Methods section.Thank you.
- Lines 33-36. Please, revise these lines since they contain identical words as lines 7-8.
Response: Thanks for the suggestion. The lines were revised.
- Lines 48-49. Please, provide a reference at the end of the sentence.
Response: Done. Thank you.
- Lines 58-61. Please, provide references supporting this information.
Response: Thanks for the suggestion. Done.
- Lines 62-63. Authors must improve this sentence. Which “above data” were analysed? Which studies?
Response: This sentence refers to the information contained in the Introduction and the studies mentioned in the Discussion.
- Line 68. Please, change “level” by “levels”.
Response: Done. Thank you.
- Line 69. “test for the worthwhile exploit of its determination”. Please, use other words.
Response: This is a good suggestion. The sentence has been corrected to make it clearer.
- According to the line 72 the study covered a population of 350 patients but, according to the information given in line 76 the study group consists of 21 pregnant women and in line 89 the authors indicate that the control is of 37 people. Authors must carefully revise the information given in these lines and provide a clear information about the number of patients included in the study, as well as the number of patients per group. What is the study group? The patients’ group with PIH? If so, please refer as the PIH group throughout the whole document.
Response: Thanks for the suggestion. The lines were revised. Throughout the document, the name of the study group was changed to PIH group.
- In the Material and Method section, authors must describe the procedure followed to statistically analyse the results obtained as wells as to evaluate the correlation between variables. Did the authors check the normality and homoscedasticity of the data? How were the AUC curve and OR values obtained?
Response: Information on statistical analysis has been added to the Material and Methods section. Thanks for the suggestion.
- In the Material and Method section, authors must describe how the Doppler examination was performed.
Response: Information on how and when the Doppler examination of uterine arteries was performer, has been added in the Material and Methods section.Thank you.
- In the Results section (lines 191-193) authors refer to obesity and diabetes? How were they diagnosed? Please, specify in the Material and Methods section.
Response: Information on how obesity and diabetes was diagnosed was included in Material and Methods section.
- Lines 92-94. How these parameters were measured. Please, add a description of the procedure followed.
Response: Information has been added in the Material and Methods section.Thank you.
- Table 1 and the content of lines 103-113 must be placed at the beginning of the Results section, subsection Cohort characteristics.
Response: Table 1 and lines 103-113 were placed at the beginning of the Results section, Cohort Characteristics subsection.
- Lines 109-110. “physical active patients alike, prophylaxis with acetyl salicylic acid and the occurrence of gestational diabetes (GDM) in the observed pregnancy”. Please, describe the procedure used to measure all these variables in the Material and Methods section.
Response: Information has been added in the Material and Methods section.Thank you.
- Line 138. “no statistical meaningful difference”. This is not a proper way to describe the results obtained. Please, improve this sentence.
Response: This sentence was improved. Thank you.
- Lines 143-144. “Figure 1. shows”. Please, delete the point.
Response: Done. Thank you.
- Lines 149-150. Please, provide an appropriate legend for Figure 1. Authors must indicate the meaning of PIH and the sample size of each group.
Response: Information about sample size of each group was mentioned. Thank you.
- Line 151. “Pronouncedly”. This is not an appropriate term. Please, indicate if the differences are either significant or not.
Response: This sentence was improved. Thank you.
- Line 151. Please correct “were shown” by “were observed”.
Response: Done. Thank you.
- Line 152. “fundamental dissimilarity differences”. Please, eliminate this expression and indicate if the differences are either significant or not.
Response: This sentence was improved. Thank you.
- Line 154. “often related”. This is quite ambiguous; authors must concretize and give a clear message based on the statistical analyses.
Response: This is a good suggestion. The sentence has been corrected to make it clearer.
- Lines 158-160. This sentence is very ambiguous. Please, provide a clear message and give P values.
Response: The p value was given. The sentence has been corrected to make it clearer. Thank you.
- Lines 160-162. Please, provide r P values confirming such a correlation. Moreover, authors must describe the procedure followed to analyse the correlation between variables in the Material and Methods section.
Response: The p value was given. Information on statistical analysis has been added to the Material and Methods section. The correlation of the variables was performed as the Spearman correlation. Thanks for the suggestion.
- Line 166. “noteworthy reciprocity”. What do the authors refer? Please, correct the whole sentence based on the results obtained from statistical analysis.
Response: The sentence has been corrected to make it clearer. Thank you.
- Line 168. “inverse correlation”. Please, indicate r and P values.
Response: The p value was given. Thank you.
- Line 173. “the results showed undesirable AUC values”. This is too vague, please provide a clear information based on both the results obtained and statistics.
Response: Clear information was provided later in the text.
- Line 177. Please, change “is” by “was”.
Response: Done. Thank you.
- Line 181. Correct “are” by “were”, and “is” by “was”.
Response: Done. Thank you.
- Authors must consider reviewing the whole Results section and use the past form.
Response: Grammatical errors were corrected and the past tense was used. The Results section has been reviewed. We believe that the results were presented in a simple and understandable way.
- Lines 183-185. The content of this lines must be placed in the Conclusion section.
Response: The content of this lines have been placed in the Conclusion section. Thank you.
- Figure 2. Please correct AUC values by using a “point” instead of a “coma”.
Response: Done. Thank you.
- Lines 187-188. Please, provide a more detailed explanation of the information given in Figure 2.
Response: The sentence has been corrected to make it clearer. Thank you.
- Line 191. “According to the study”. Which study do the authors refer?
Response: The sentence has been corrected to make it clearer. Thank you.
- Lines 191-193. “the presence of PIH in the previous pregnancy”. This is quite confusing considering that the authors did not refer to “previous pregnancies” of the patients included in the study. Therefore, a deep revision is much warranted.
Response: Information on the prevalence of PIH in previous pregnancy is presented in Table 1.
- Lines 194-196. The content of these lines must be placed in the Conclusions section.
Response: The content of this lines have been placed in the Conclusion section. Thank you.
- Line 198. Please, change “occurs” by “occurred”.
Response: Done. Thank you.
- Line 199. Please, change “are” by “were”.
Response: Done. Thank you.
- Line 200. Please change “does not” by “did not”.
Response: Done. Thank you.
- Authors must revise the whole Result section and provide a clear message. Moreover, this section must be written in past tense.
Response: Grammatical errors were corrected and the past tense was used. The Results section has been reviewed. We believe that the results were presented in a simple and understandable way.
- Line 205. “Much higher values”. What do the authors refer? Please, provide a clear message by giving objective data.
Response: Objective data was given. Thank you.
- Line 208. Please, change “is” by “was”.
Response: Done. Thank you.
- Lines 209-211. The content of these lines must be placed in the Conclusions section.
Response: The content of this lines have been placed in the Conclusion section. Thank you.
- Lines 216-225. Subsection 4.1. Summary. I can not see the relevance of this subsection considering that most of the information it contains is already given in the Introduction section.
Response: This section has been deleted. Thank you.
- Lines 227-310. Subsection 4.2. Comparison with Existing Literature. Please, eliminate this subsection title. Moreover, authors must revise the information given in these lines by analysing and comparing their results with those obtained in similar approaches instead of summarizing the research conducted in other studies.
Response: Subsection title was deleted. It was revised the information given in these lines by analyzing and comparing their results with those obtained in similar approaches. Thanks for the suggestion.
- Lines 317-320. This is not a limitation of your study. Authors must start the Discussion section by highlighting the low number of studies related to this subject and the great diversity of experimental approaches.
Response: Discussion section was started by highlighting the low number of studies related to this subject and the great diversity of experimental approaches. Thank you.
- Lines 321-333. Subsection 4.4. Implications for Research and Clinical Practice. The information given in this section is a mere repetition of the information given in previous sections of the manuscript.
Response: This section has been deleted. Thank you.
- Lines 335-341. The content of these lines does not provide any new information. This section must contain the most relevant findings of the research conducted.
Response: This lines have been deleted. Thank you.
- Lines 342-349. Please, avoid repeating literally the results obtained, and use new words. Authors must highlight that the sample size is reduced and so their results are preliminary.
Response: Authors used new words for giving information. It was highlight that the sample size is reduced and so their results are preliminary.
- The reference list is quite dated. Authors must cite recent publication to support their findings.
Response: The reference list has been updated. Thank you.
Round 2
Reviewer 2 Report
The authors have introduced all the modifications suggested by the reviewers, but the manuscript still needs improvement before its acceptance for publications. Moreover, I strongly encourage the authors to carefully review English grammar.
- Lines 8-9. “The etiology of PIH has not been clearly defined, and impairing effective prophylaxis and treatment”. Please, revise the content of these lines and provide a clear message.
- Line 14. “Blood samples were collected and the frozen at three time points”. Please, eliminate “the” before “frozen”. On the other hand, I am not sure that the blood samples could be frozen at three different points. I suppose that authors refer that blood samples were collected at three different time points and then frozen. If so, please revise the sentence and provide a clear message. Moreover, according to lines 15 and 16, these three different points correspond to a different trimester of gestation. I strongly encourage the authors to carefully revise the content of lines 14-16, integrate the information given and provide a clear message.
- Line 15. “Copeptin levels”. Please indicate the measure units in brackets.
- Line 23. Please, correct “In conclusion:” by “In conclusion,”.
- Lines 33-39. Please, provide bibliographic references.
- Materia and Method section. Please, provide bibliographic references supporting the methodologic approach and procedures used.
- Lines 74-77. Please, revise the content of these lines and provide a clear message.
- Lines 87-88. Please, revise the content of these lines and provide a clear message.
- Line 149. Please, refer as “statistical analysis” instead of “statistical research”.
- Line 186. “There were no particular differences”. Please, note that this is not a correct form to indicate that the differences between both groups were not statistically significant. Please, revised the whole sentence and provide an appropriate message from a scientific point of view.
- Lines 188-191. Authors must revise the content of these lines by indicating that the differences were not statistically significant.
- Line 191. Please remove the point after “Table 1.”.
- Line 212. Please, refer as “was significantly higher” instead of “was comparatively higher”.
- Line 219. “The sampling time in the first and second trimesters did not differ significantly”. I am unable to understand the meaning of this sentence; please, provide a clear message.
- Lines 226-228. Please, revise the content of these lines and provide a clear message.
- Lines 263-264. “Lower, but also high risk occurred in pregnant women with high levels of copeptin and diabetes”. The content of these lines could be quite confusing; please, provide a clear message.
- Line 265. Please, correct “are” by “were”.
- Line 279. Please, correct “limited number” by “reduced number”.
- Lines 283-285. Please, be careful with content of these lines that it seems contradictory to the content of lines 279-280.
- The second paragraph of the Discussion needs deep revision. Authors describe the results of other researchers without providing any about their biological significance as compared to their own results. Moreover, this paragraph is excessively long, and it needs a deep redistribution of the information given in order to provide a clear message.
- Line 286. “the results of some of them are not convincing”. Authors must argue this affirmation, by indicating which results are “not convincing” and why. Please, also not the expression “not convincing” is not appropriate in this context.
- Lines “the size of the studied groups was small, which may also raise doubts about their objectivity”. This is a hard affirmation; please prove arguments that support it and try to use other terms to express your disagreement with previous research.
- Lines 293-295. Please, provide a bibliographic reference at the end of the sentence.
- Lines 307-308. “We have come to a different conclusion in our study”. Please, note that the content of these lines is repeated in lines 319-320.
- Lines 311-319. The authors provide some results, but I am not sure if they are their own results, or the results obtained from other authors. In the further, which is the point of describing the results of other researchers? I think that it could be more interesting to analyse the biological significance of the differences between their own results and the results obtained by other researchers.
- Lines 339-365. The third paragraph of the Discussion also need a deep revision to provide a clear message but not just to cite the results obtained by other authors.
Author Response
Response to Reviewer 3 Comments
The authors have introduced all the modifications suggested by the reviewers, but the manuscript still needs improvement before its acceptance for publications. Moreover, I strongly encourage the authors to carefully review English grammar.
We thank this reviewer for the positive evaluation of our manuscript and critical comments that has helped the revised version.
- Lines 8-9. “The etiology of PIH has not been clearly defined, and impairing effective prophylaxis and treatment”. Please, revise the content of these lines and provide a clear message.
Response: The content of these lines was revised. We tried to get a clear message. Thank you.
- Line 14. “Blood samples were collected and the frozen at three time points”. Please, eliminate “the” before “frozen”. On the other hand, I am not sure that the blood samples could be frozen at three different points. I suppose that authors refer that blood samples were collected at three different time points and then frozen. If so, please revise the sentence and provide a clear message. Moreover, according to lines 15 and 16, these three different points correspond to a different trimester of gestation. I strongly encourage the authors to carefully revise the content of lines 14-16, integrate the information given and provide a clear message.
Response: Thanks for the suggestion. The sentence has been corrected to make it clearer.
- Line 15. “Copeptin levels”. Please indicate the measure units in brackets.
Response: Done. Thank You.
- Line 23. Please, correct “In conclusion:” by “In conclusion,”.
Response: Done. Thank You.
- Lines 33-39. Please, provide bibliographic references.
Response: Bibliographic references was provided. Thanks for the suggestion.
- Materia and Method section. Please, provide bibliographic references supporting the methodologic approach and procedures used.
Response: Bibliographic references was provided. Thanks for the suggestion
- Lines 74-77. Please, revise the content of these lines and provide a clear message.
Response: The content of these lines was revised. We tried to get a clear message. Thank you.
- Lines 87-88. Please, revise the content of these lines and provide a clear message.
Response: The content of these lines was revised. We tried to get a clear message. Thank you.
- Line 149. Please, refer as “statistical analysis” instead of “statistical research”.
Response: Thank you. Done.
- Line 186. “There were no particular differences”. Please, note that this is not a correct form to indicate that the differences between both groups were not statistically significant. Please, revised the whole sentence and provide an appropriate message from a scientific point of view.
Response: The content of these lines was revised. We tried to get a clear message. Thank you.
- Lines 188-191. Authors must revise the content of these lines by indicating that the differences were not statistically significant.
Response: The content of these lines was revised. We tried to get a clear message. Thank you.
- Line 191. Please remove the point after “Table 1.”.
Response: Done. Thank you.
- Line 212. Please, refer as “was significantly higher” instead of “was comparatively higher”.
Response: Done. Thank you.
- Line 219. “The sampling time in the first and second trimesters did not differ significantly”. I am unable to understand the meaning of this sentence; please, provide a clear message.
Response: The content of these lines was revised. We tried to get a clear message. Thank you.
- Lines 226-228. Please, revise the content of these lines and provide a clear message.
Response: Thanks for the suggestion. The sentence has been corrected to make it clearer.
- Lines 263-264. “Lower, but also high risk occurred in pregnant women with high levels of copeptin and diabetes”. The content of these lines could be quite confusing; please, provide a clear message.
Response: The content of these lines was revised. We tried to get a clear message. Thank you.
- Line 265. Please, correct “are” by “were”.
Response: Done. Thank you.
- Line 279. Please, correct “limited number” by “reduced number”.
Response: Done. Thank you.
- Lines 283-285. Please, be careful with content of these lines that it seems contradictory to the content of lines 279-280.
Response: The content of these lines was revised. We emphasized that few reports of copeptin levels in PIH patients have been published compared to a significant number of preeclampsia studies. Thank you.
- The second paragraph of the Discussion needs deep revision. Authors describe the results of other researchers without providing any about their biological significance as compared to their own results. Moreover, this paragraph is excessively long, and it needs a deep redistribution of the information given in order to provide a clear message.
Response: The third paragraph of the Discussion was revised. Thank you.
- Line 286. “the results of some of them are not convincing”. Authors must argue this affirmation, by indicating which results are “not convincing” and why. Please, also not the expression “not convincing” is not appropriate in this context.
Response: Upon reflection, these lines were removed. Thanks for the suggestion.
- Lines “the size of the studied groups was small, which may also raise doubts about their objectivity”. This is a hard affirmation; please prove arguments that support it and try to use other terms to express your disagreement with previous research.
Response: Thanks for the suggestion. The content of these lines was revised.
- Lines 293-295. Please, provide a bibliographic reference at the end of the sentence.
Response: Bibliographic references was provided. Thanks for the suggestion.
- Lines 307-308. “We have come to a different conclusion in our study”. Please, note that the content of these lines is repeated in lines 319-320.
Response: We have changed the content of this line. Thank you.
- Lines 311-319. The authors provide some results, but I am not sure if they are their own results, or the results obtained from other authors. In the further, which is the point of describing the results of other researchers? I think that it could be more interesting to analyse the biological significance of the differences between their own results and the results obtained by other researchers.
Response: We have changed the content of this line. The Discussion was revised. Thank you.
- Lines 339-365. The third paragraph of the Discussion also need a deep revision to provide a clear message but not just to cite the results obtained by other authors.
Response: The third paragraph of the Discussion was revised. Thank you.